# A unitary mechanism underlies adaptation to both local and global environmental statistics in time perception

Tianhe Wang[1,2]*, Yingrui Luo[1], Richard B. Ivry[2], Jonathan S. Tsay[2], Ernst Pöppel[1,3], Yan Bao[1,3,4]*

1 School of Psychological and Cognitive Sciences, Peking University, Beijing, China, 2 Department of Psychology and Helen Wills Neuroscience Institute, University of California Berkeley, Berkeley, California, United States of America, 3 Institute of Medical Psychology, Ludwig Maximilian University, Munich, Germany, 4 Beijing Key Laboratory of Behavior and Mental Health, Peking University, Beijing, China

* tianhewang@berkeley.edu (TW); baoyan@pku.edu.cn (YB)

## Abstract

Our duration estimation flexibly adapts to the statistical properties of the temporal context. Humans and non-human species exhibit a perceptual bias towards the mean of durations previously observed as well as serial dependence, a perceptual bias towards the duration of recently processed events. Here we asked whether those two phenomena arise from a unitary mechanism or reflect the operation of two distinct systems that adapt separately to the global and local statistics of the environment. We employed a set of duration reproduction tasks in which the target duration was sampled from distributions with different variances and means. The central tendency and serial dependence biases were jointly modulated by the range and the variance of the prior, and these effects were well-captured by a unitary mechanism model in which temporal expectancies are updated after each trial based on perceptual observations. Alternative models that assume separate mechanisms for global and local contextual effects failed to capture the empirical results.

**Data Availability Statement:** Data for this paper and codes for analyses are available at https://osf.io/2dz7k/.

## Author summary

Our perceptual system can actively adapt to the statistical properties of the environment in multiple time scales. For example, the perceived duration of an event is biased by the mean duration of events observed in a relatively long period and also by the durations of recently processed events. Here we ask whether these two effects reflect the operation of separate mechanisms or a unitary mechanism. We develop a series of computational models of independent and unitary mechanisms, and use experimental manipulations that generate predictions which allow us to evaluate the models. We show that serial dependence, the signature of short-term adaptation, is modulated by the long-term context. The results are consistent with the predictions of a unitary mechanism model which assumes the global prior is updated in a trial-by-trial manner. The alternative models that assume separate mechanisms fail to capture the empirical results. These results provide a

**Funding:** This work was supported by the National Natural Science Foundation of China (31771213 and 31371018) to Y.B. The funders had no role in study design, data collection and analysis, decision to publish, or preparation of the manuscript.

**Competing interests:** I have read the journal's policy and the authors of this manuscript have the following competing interests: RI is a co-founder with equity in Magnetic Tides, Inc.

comprehensive picture of how the timing system jointly adapts to short- and long-term environmental statistics.

## Introduction

The internal representation of temporal information is essential for a wide range of cognitive functions, from anticipating future events to controlling movements [1–3]. To improve the precision of temporal perception, the timing system flexibly adapts to the statistical properties of the current context [4, 5]. For example, when presented with a set of durations in a perception task, participants have a strong tendency to overestimate relatively short durations and underestimate relatively long durations. Thus, the perceived duration is biased toward the mean of the set [4, 6–9], a phenomenon known as the "central tendency effect." This effect indicates that temporal perception is sensitive to the global temporal context.

It has also been shown that temporal perception can adapt on a rapid timescale. Participants' perception of the current duration is attracted toward the duration of the previous stimulus [10–12]. That is, the duration is perceived to be longer after a relatively long duration stimulus compared to a relatively short duration stimulus [13–16]. This phenomenon, known as serial dependence, suggests that the perceptual system is also sensitive to the local statistics of the environment [17, 18].

Central tendency and serial dependence effects have been observed across a wide range of perceptual tasks [17–22], indicating that they reflect general principles of how the perception system adapts to the statistics of the environment. Both phenomena can be explained under a Bayesian framework [23]. On the one hand, the observers appear to construct a relatively stable global prior that reflects the distribution of the stimulus set [6, 24, 25]. Following Bayesian integration, the current perception is biased toward the mean of the global prior, the central tendency effect. On the other hand, the observer also appears to build a temporal expectancy based on the most recent stimulus, inducing a bias in judging the duration of the current stimulus towards recently experienced stimuli, the serial dependence effect [26].

Although the central tendency and serial dependence effects describe two ways in which context can influence behavior, it remains unclear whether they reflect the operation of a unitary process or two separate, adaptive processes. From a unitary view, the perceptual system continuously updates the global prior based on new observations, and the trial-by-trial updating of the global prior could influence the subsequent perception, giving rise to serial dependence [10, 11, 27]. Alternatively, there may be two adaptive systems that operate on different timescales in response to environmental statistics, building up global *and* local priors that give rise to central tendency and serial dependence, respectively.

To arbitrate between these hypotheses, we used a temporal reproduction task [6, 28–30] in which participants reproduce an interval specified by a visual stimulus. Across conditions, we manipulated the global distribution by sampling the target durations from different temporal distributions. If serial dependence and central tendency arise from a shared mechanism, the magnitude of the serial dependence effect would be impacted by the global temporal distribution. Alternatively, if serial dependence and central tendency arise from distinct mechanisms, the magnitude of the serial dependence effect would remain invariant across a wide range of global temporal distributions. We formalized these hypotheses and compared the predictions of these computational models with the empirical results. By combining our behavioral experiments and model-based analyses, we sought to unravel the computational mechanisms underlying the influence of context on temporal perception.

## Results

### Serial dependence in time perception is attractive and non-linear

We begin this project by examining serial dependence and central tendency simultaneously in a widely employed temporal reproduction task, the "ready-set-go" task [6]. Participants observed a pair of visual events that defined a target interval ("Ready" and "Set") and then made a single button press ("Go"), attempting to produce an interval between the Set and Go signals that reproduced the target interval (Fig 1A). The target durations were randomly sampled from a uniform distribution that ranged from 500 to 900ms.

The reproduced durations exhibited robust regression towards the mean (Fig 2A), replicating the central tendency effect seen in previous studies [10, 29, 31]. Quantitatively, the slope of the reproduced duration to the target duration was significantly smaller than one (0.65 ± 0.20; t(11) = -6.08; p<0.001, S1A Fig). To examine serial dependence, we first calculated a "deviation" index, the difference between the reproduction of a target duration on a given trial and the individual's mean reproduction to that target duration across all trials (see Fig 2A). The serial dependence effect is calculated by the change in the deviation index as a function of the difference between the previous and the current target durations. This method minimizes potential artifacts in the serial dependence function that may be induced by regression to the mean and reproduction biases [32, 33].

We found that the deviation is biased towards the previous stimulus (Fig 2B): When the target duration on trial n-1 was longer than the stimulus on trial n, the reproduced duration tended to be longer than average, and vice-versa. This indicates that the reproduction on the current trial is attracted towards the stimulus duration (or reproduced duration) of the previous trial. Notably, the shape of serial dependence is non-linear: The attraction effect peaks when the current stimulus differs from the previous stimulus by approximately 100ms and then falls off when the difference grows larger.

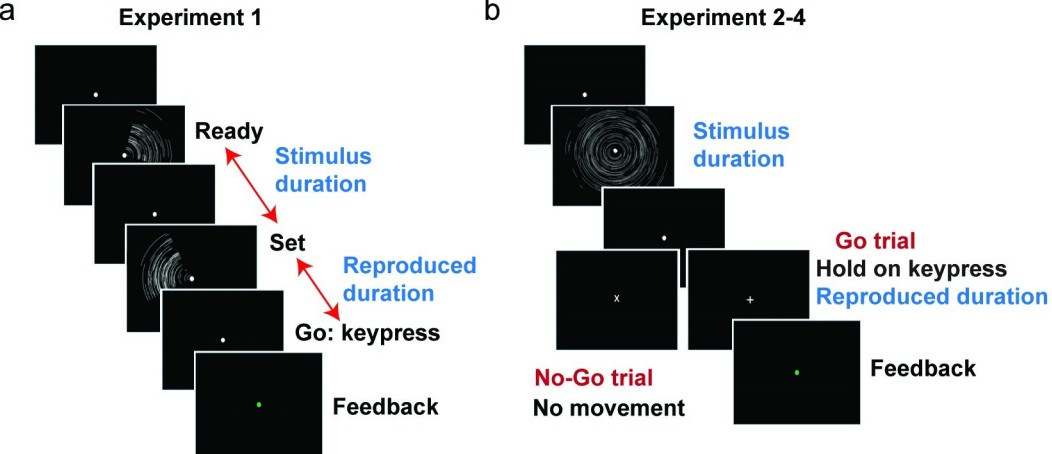

**Fig 1. Trial and task structure.** (a) The stimulus sequence used in Experiment 1 ("ready-set-go" task). Two 100-ms stimuli flashed in sequence, signifying first a "Ready" signal and then a "Set" signal; the target duration was the interval between the onset times of the "Ready" and "Set" signals. Participants were instructed to press the space bar to reproduce the temporal interval after the "Set" signal. Performance feedback was conveyed for 50 ms via the color of the fixation cross (green = correct; red = incorrect). (b) The stimulus sequence used in Experiments 2–4. We used a duration reproduction task including both go and no-Go trials. A ripple-shaped stimulus was presented for a fixed duration denoting the temporal interval. After a 300 ms interval, the fixation point became either a "+" sign or "x" sign, signaling either a "go" or "no-go" trial, respectively. In Go trials, participants reproduced the temporal interval by holding down a key. When the key was released, the fixation point turned to a grey circle signaling the end of the trial. In the no-Go trial, participants were asked to withhold any movement and fixate until the "x" switched to a grey circle after 700ms.

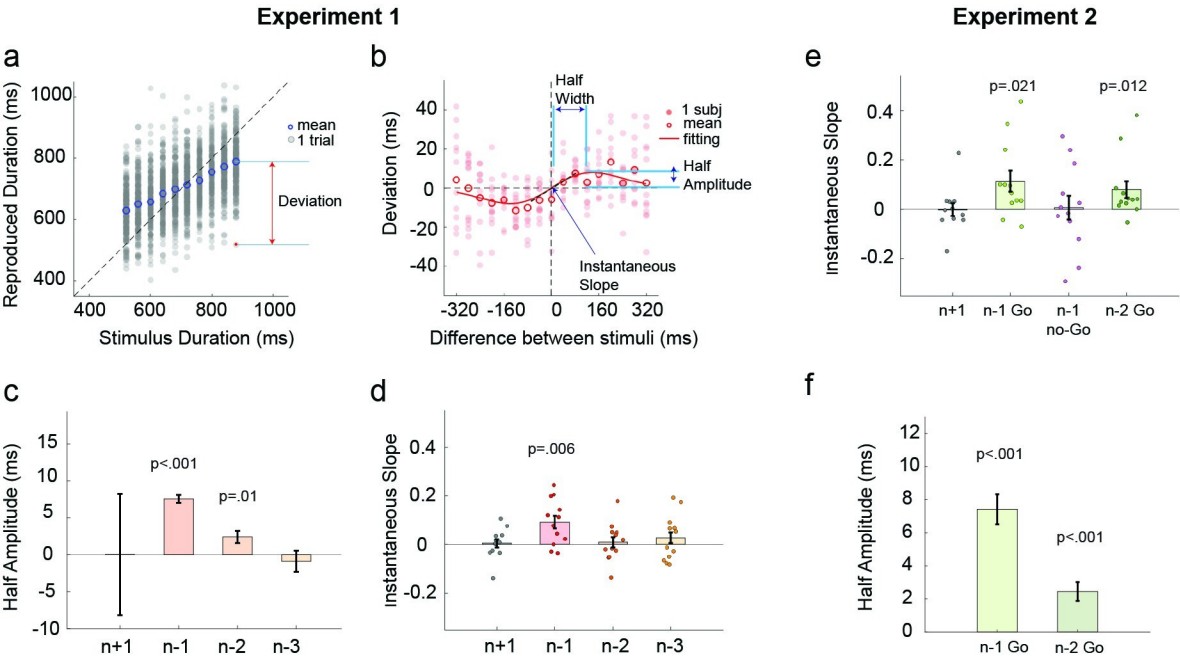

**Fig 2. Serial dependence in duration reproduction.** Experiment 1: (a) Reproduced durations of a representative participant. Grey dots represent the reproduced duration of each trial. The blue circle represents the participant's mean reproduced duration for each target duration. (b) Serial dependence is evident in the non-linear relationship between the deviation index, the current reproduced duration minus the mean reproduced duration, as a function of the temporal difference between the previous target duration and the current target duration (positive values indicate the previous target was longer). Filled dots denote individual participants. Empty circles denote the mean of all 12 participants. The red solid line represents the best-fitted Derivative of Gaussian (DoG) model at the group level. (c) Half-amplitude of the best-fitted DoG in Experiment 1. Error bars represent an estimate of the standard error obtained from jackknife resampling. (d) Instantaneous slope obtained from the fitted DoG curve for trials n-1, n-2, n-3, and n+1 trials. Experiment 2: (e) Instantaneous slope obtained from the fitted DoG curve with respect to n-1 Go, n-1 no-Go trials, n-2 Go trial (when n-1 is no-Go trials), and n+1 (control condition). (f) Half-amplitude of the best-fitted DoG for trials n-1 and n-2.

This non-linear function is well captured by a derivative of Gaussian (DoG) curve [17, 32]. To quantify how previous stimuli influence the current perception in experiment 1, we fitted a DoG to the function of the difference between the target duration in trial n and trial 1-back, 2-back, or 3-back, respectively. We used the instantaneous slope at zero to measure the sign of serial dependence (see Fig 2B). Positive instantaneous slopes indicate that the current perception is attracted towards the previous stimulus duration; negative slopes indicate that the current perception is repelled from the previous stimulus duration. We found a positive slope when calculating serial dependence based on trial n-1 ($0.092 \pm 0.094$, $t(11) = 3.39$, $p = 0.006$; Fig 2D), but not when the calculation was based on trial n-2 ($0.009 \pm 0.079$, $t(11) = 0.39$, $p = 0.70$) or n-3 ($0.028 \pm 0.095$, $t(11) = 1.00$, $p = 0.34$). To better estimate the magnitude of the serial dependence effect in response to the previous stimuli, we fit a DoG at the group level. The DoG provided a good fit (Fig 2B), outperforming the null-model ($\Delta AIC_n = -46.3 \pm 7.7$) and linear-model ($\Delta AIC_l = -18.6 \pm 4.6$) for the trial n-1 function. The half-amplitude of this function is $7.6 \pm 0.6$ms, and the half-width is $95.8 \pm 10.1$ms (Fig 2C). When analyzed at the group level, we also get a significant serial dependence effect from trial n-2 ($2.4 \pm 0.8$ ms, $z = 3.00$, $p = 0.001$), but not from trial n-3 (Fig 2C), indicating there might be a weak attractive effect that is not evident in the instantaneous slopes calculated at the individual level. We return to this issue below.

## Serial dependence mainly originates from the temporal reproduction

In Experiment 2, we asked whether the serial dependence effect originates from temporal perception or temporal reproduction. To test this, we included No-Go trials in which participants were instructed not to respond. Temporal reproductions following Go trials could reflect biases arising from processes associated only with perception, only with motor production, or both. In contrast, temporal reproductions following No-Go trials should only be influenced by a perceptual bias. Importantly, the Go/No-Go signal was only presented after the target duration, ensuring that participants encoded the target duration on both Go and No-Go trials (Fig 1B).

In trials immediately following Go trials, we replicated the serial dependence effect observed in experiment 1 (instantaneous slope: 0.114 ± 0.042; Wilcoxon test: z = 2.03, p = 0.021; Fig 2E) with a half-amplitude of 7.4 ± 0.9ms (Fig 2F) and half-width of 111.0 ± 15.6ms (S3A Fig). In contrast, we did not find a serial dependence effect from the previous stimulus following a No-Go trial (instantaneous slope: 0.006 ± 0.049; Wilcoxon test: z = 0.01, p = 0.97; Figs 2E and S3B). This dissociation supports the idea that serial dependence in timing largely arises from factors associated with temporal reproduction. We recognize that this may reflect how reproduction is influenced by attention, decision making, memory, or the response itself. Consistent with this notion, when we examined trial triplets composed of Go-NoGo-Go trials, we found a serial dependence effect on the second Go trial towards the first Go trial (rather than the intervening NoGo trial; instantaneous slope, 0.080 ± 0.034, Wilcoxon test: z = 2.25, p = 0.012; Fig 2E–2F; S3C Fig).

## Modeling central tendency and serial dependence

Having established robust signatures of the central tendency and serial dependence effects, we now ask whether these two effects are generated by a unitary mechanism or reflect separate mechanisms that are shaped by global and local statistics, respectively. To test this, we formalized our intuitions into a series of computational models.

To explain the central tendency effect, we begin with a Bayesian-least-square model [6] that assumes the observer's global temporal expectancy is static (i.e., global prior), with bias constrained only by the global distribution of temporal stimuli (Global-Only model, Fig 3A and 3B). The observer's temporal estimation on a given trial is the Bayesian integration of the global prior with the actual target duration (corresponding to the likelihood in Bayesian theory). As such, the estimated duration of each trial is biased towards the global prior, giving rise to the central tendency effect (Fig 4A). However, given that the Global-Only model uses a fixed prior, the observer's estimate will not be influenced by the local context (e.g., recently experienced stimuli). That is, the basic Bayesian model cannot account for the serial dependence effect (Fig 4B).

We next considered a unitary mechanism Bayesian Interference model that assumes that the observer integrates the likelihood and a local prior based solely on the local context (Local-Only model, Fig 3E and 3F). The weights of the local prior and likelihood are determined by the difference between distributions associated with the current sample and recently experienced samples. Specifically, the observer will rely less on the local prior when it is far away from the likelihood and more on the local prior when it is close to the likelihood. The Local-Only model can produce a non-linear serial dependence effect (Fig 4B). However, this model cannot simultaneously capture the observed serial dependence and central tendency effects (Figs 4A–4C and S4). Parameter values that generate the observed serial dependence effect produce a smaller than observed central tendency effect. Parameter values that generate the observed central tendency effect produce an unreasonably large serial dependence effect.

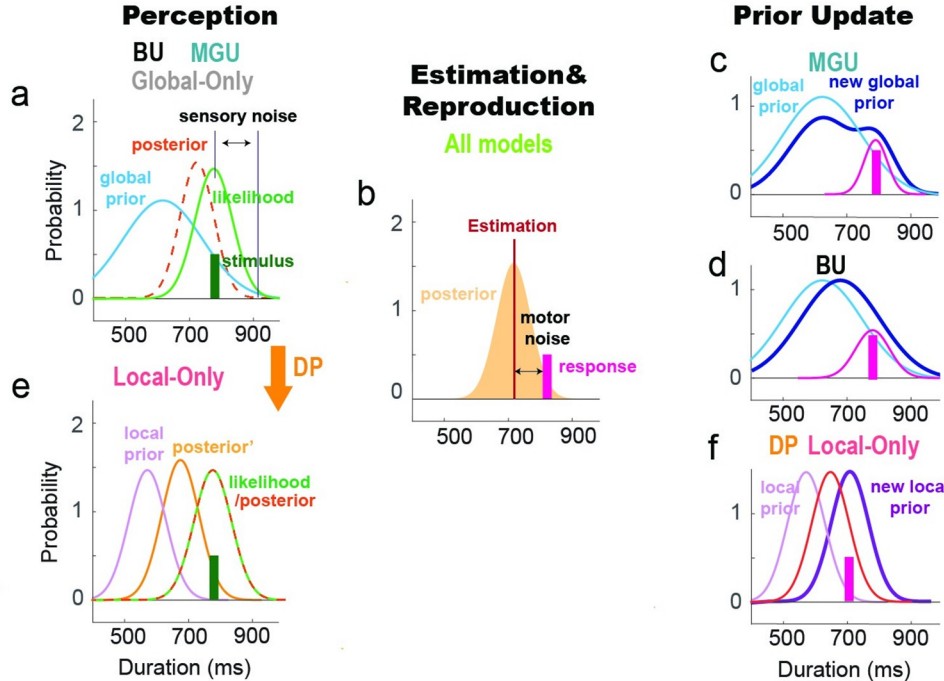

**Fig 3. Schematics of the different models.** (a) In all of the models, the target duration (green) is represented as a normal distribution (likelihood) centered at the target duration with perceptual noise. The global prior (blue) is based on the distribution of the target stimuli. The posterior (yellow) is obtained by multiplying the global prior by the likelihood. (b) The mean of the posterior is computed as a single-value estimate. The reproduced duration is sampled from a normal distribution centered on the estimate with motor noise. (c-d) In the BU and MGU models, the observer updates the prior based on the reproduced duration after the motor response. (c) For the MGU model, a scaled normal distribution centered at the reproduced duration is added to the old prior, and the new prior is normalized. (d) For the BU model, the prior is updated with a Kalman filter. (e) For the Local-Only model, the likelihood is integrated with the local prior in a non-linear manner, with the weight on the local prior increasing when the likelihood and local prior are similar and decreasing when they are dissimilar. In the DP model, the observer is assumed to hold two priors, one global and one local. After the posterior is computed based on the global prior and likelihood, the posterior is integrated with the local prior to generate a second posterior (posterior'). (f) In the second step of the DP and Local-Only models, the new local prior is fully determined by the motor response of the current trial. As with the other three models, the reproduced duration is drawn from a normal distribution centered at the mean of the posterior with motor noise.

Thus, a local prior by itself is not sufficient to explain adaptative behavior in the current experiments, indicating that the central tendency effect is not a by-product of serial dependence.

We then considered models capable of simultaneously generating serial dependence and central tendency effects. First, we considered a unitary mechanism model in which the global prior is updated across trials (Bayes-Updating model, BU). Given that serial dependence is driven by temporal reproduction (Exp 2), we assume that, following Bayes rule, the prior is updated by integrating a Gaussian centered at the reproduced duration. Since the priors and posteriors are both Gaussians, the Bayesian integration can be simplified into a linear weighted sum of the means (i.e., a Kalman filter, Fig 3D). This model is mathematically similar to what has been previously described as an internal reference model [10, 11, 26, 34, 35]. While this model predicts a central tendency effect and an attractive serial dependence effect, the predicted serial dependence function is near-linear (Fig 4B), inconsistent with the empirical results of Exps 1–2.

To get a non-linear serial dependence function, the prior can be updated in a non-Bayesian manner. We assume that the brain creates the prior distribution by summing multiple

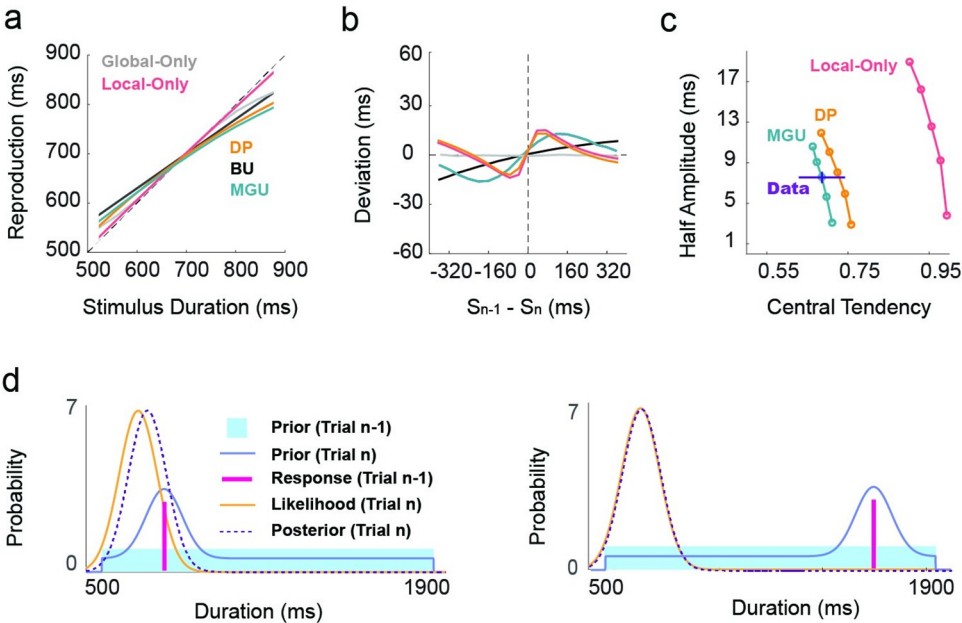

**Fig 4. Model simulations.** Predicted central tendency (a) and serial dependence (b) effects for the five models. (c) Predicted relationship between central tendency and serial dependence for the models. Each line is the prediction of one model, and each dot is the prediction with a specific learning rate parameter. The half-amplitude increases with the learning rate. The blue "+" represents the empirical data. The length of the bars indicates SE. (d) Illustration of how the MGU model generates a non-linear serial dependence. For illustration purposes, we depict the n-1 prior as a broad, uniform distribution ranging from 500-1900ms, the range used for the long condition in Exp 3. When the current and previous durations are close (left), the previous duration can influence the shape of the prior in the range around the current duration and make the current posterior shift more relative to the likelihood. In this region, the bias of the posterior increases with the distance between two successive stimuli. However, when the current and previous duration are distinct (Right), the previous duration cannot influence the shape of the prior around the likelihood, and thus, the attraction effect decreases.

Gaussian distributions with different weights (Mixed Gaussian Updating model, MGU). When perceiving a new duration, the prior is updated by adding a Gaussian centered at the reproduced duration to the old prior, followed by normalization (Fig 3C). This kind of computation has previously been suggested to account for how Purkinje cells in the cerebellum adapt to the prior for representing temporal information [30]. Since the global prior is updated locally, the attraction effect from trial n-1 decreases when the duration of the current stimulus is far from the duration of the n-1 reproduction (Fig 4D). Thus, this model predicts a non-linear serial dependence function and serial dependence in accordance with the empirical results in experiments 1–2 (Fig 4A–4C).

In contrast to the unitary mechanism models described above, an alternative way to produce both serial dependence and central tendency effect is to consider a hybrid model in which the two biases arise from two distinct processes (Dual Priors model, DP). Specifically, the DP model assumes that the current stimulus (likelihood) is integrated with a static global prior (Fig 3A), generating central tendency. The posterior is then integrated with a local prior (Fig 3F), inducing non-linear serial dependence. As such, the Dual Priors model can predict both the central tendency and non-linear serial dependence effects observed in the data (Fig 4A–4C). Note that these two effects are generated by two mechanisms that adapt separately to the global and local temporal context.

In sum, our computational analyses point to two candidate models that can account for the observed central tendency and serial dependence effects. These two models diverge in that the

MGU model postulates that central tendency and serial dependence effects arise from a unitary mechanism (non-linear updates to a global prior). In contrast, the DP model postulates that these two effects arise from separate mechanisms (a fixed global prior paired with updates to a local prior). In the following sections, we sought to arbitrate between the MGU and the DP models.

## Contextual effect on serial dependence

A key difference between the MGU and DP models is that the latter assumes the local prior is mainly determined by the duration of trial n-1; as such, a serial dependence effect from the n-2 trial will be very weak. In contrast, the MGU predicts a robust serial dependence effect from trial n-2, given that all (recent) observations are integrated into the prior. In experiment 1, the group level analysis indicated a positive serial dependence effect from the n-2 trial. However, this effect was not observed in the individual analyses of the instantaneous slope values.

To address this discrepancy, we sought to enhance the serial dependence effect and examine whether it will be manifest beyond trial n-1. The MGU and DP models both predict that extending the range of the durations in the test set will enhance the trial n-1 serial dependence effect (Fig 5). In the DP, sensory noise scales with duration, a form of Weber's law. Because of this, the likelihood becomes relatively flat when the range is increased (Fig 5E). This will result in a greater influence of the previous production and, thus, a strengthening of the serial dependence effect (Fig 5F–5G). Sensory noise also influences the serial dependence effect in a similar way in the MGU. In addition, the MGU model postulates that the observer builds a concentrated prior when the range is limited (Fig 5D right). As such, the prior is resistant to updating, yielding a weak serial dependence effect. When the range is expanded, the prior becomes more distributed (Fig 5D left), resulting in a more pronounced local change after each update, and thus, a stronger serial dependence effect (Fig 5B and 5C). Importantly, as noted above, the MGU and DP models generate very different predictions regarding concerning a trial n-2 serial dependence effect (Fig 6A): The MGU predicts that the serial dependence effect should be observed from trial n-2, with a half-amplitude slightly attenuated relative to trial n-1, whereas the DP predicts almost no serial dependence effect from trial n-2.

Given the predictions of the two models, we extended the range of the target durations in experiment 3. We applied two test sets, one ranging from 520-1260ms (Medium range) and the other from 560-1860ms (Large range, Fig 5A). We compared the results with those obtained in Experiment 1, in which the target durations ranged from 520-880ms (Short range). Extending the range of target duration successfully enhanced the serial dependence effect from trial n-1. The best-fitted DoG of the medium condition ($\Delta AIC_n$ = -27.5 ± 4.2; $\Delta AIC_l$ = -10.8 ± 2.2) and long condition ($\Delta AIC_n$ = -20.7 ± 4.4; $\Delta AIC_l$ = -10.0 ± 2.1) had a higher peak and was broader than that for the short condition (Fig 5H). Correspondingly, the half-amplitude increased as the distribution became wider (Fig 5I; medium: 11.8 ± 0.9 ms; long: 22.6 ± 2.0 ms; Zs > 4.9, ps < 0.001). Similarly, the half-widths also increased as the distribution became wider (Zs > 10.5, ps < 0.001), indicating an influence from more distant productions on the previous trial when the test set range increased.

The key question in Experiment 3 centers on the trial n-2 data: Will the increase in the magnitude of the serial dependence effect from trial n-1 be accompanied by a stronger serial dependence effect from non-adjacent trials? In the analysis of the individual functions, there was a significant positive instantaneous slope for the trial n-2 data in both the medium and long conditions (Fig 6B). Trial n-3 also showed a tendency for a positive bias, although this did not reach significance. Moreover, fitting the DoG at the group level showed a significant positive half-amplitude for the serial dependence function from trial n-2 (Fig 6C). Thus, the results

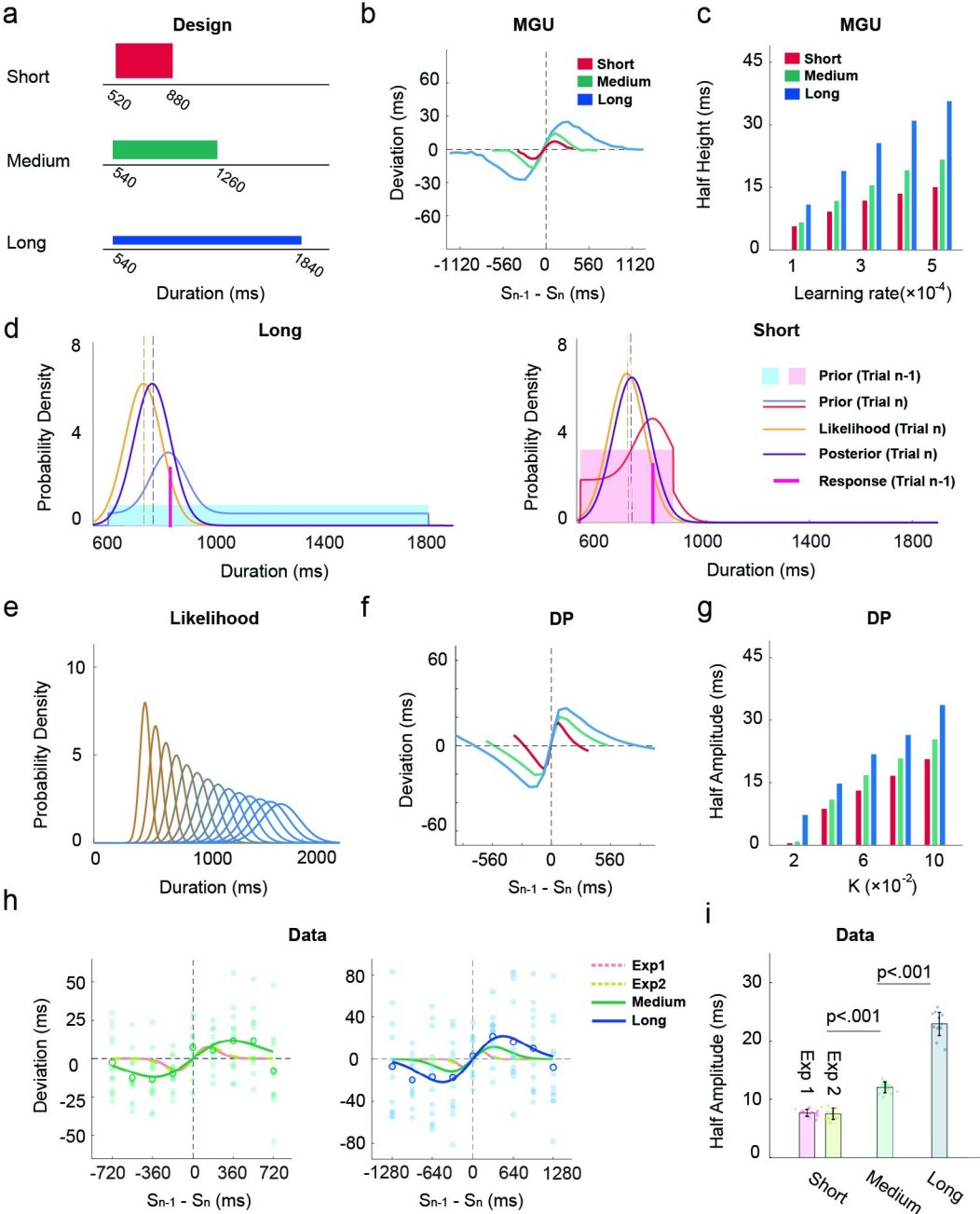

**Fig 5. The range of the target duration distribution influences the serial dependence effect.** (a) Distributions of target durations in different contexts. (b-c) Simulations of the MGU model predict that the half-amplitude of the serial dependence effect will increase as the range of target duration is increased. (d) In the MGU model, changing the range of the target durations will impact the width of the prior and, therefore, the serial dependence effect. (e) In the PD model, since the scalar property in time perception such that the ratio between the SD and mean is a constant, the likelihood will become flatter as the target duration increases. (f-g) Simulations of the DP model also predict that the half-amplitude of serial dependence increases substantially as the range of target duration is increased. (h) Serial dependence effects for trial n-1 in the medium (left) and long condition (right). Filled dots represent individual participants. Blank circles represent the average of all participants. The turquoise and blue lines represent the best-fitted DoG in the medium and long conditions, respectively. The red and green dash lines represent the best-fitted DoG in Experiments 1 and 2, respectively. (i) Half-amplitude of the best-fitted DoG for serial dependence effect from trial n-1 in Experiments 1, 2, and 3 (medium and long conditions). Each filled dot represents an estimate from jackknife resampling. Error bars represent standard error.

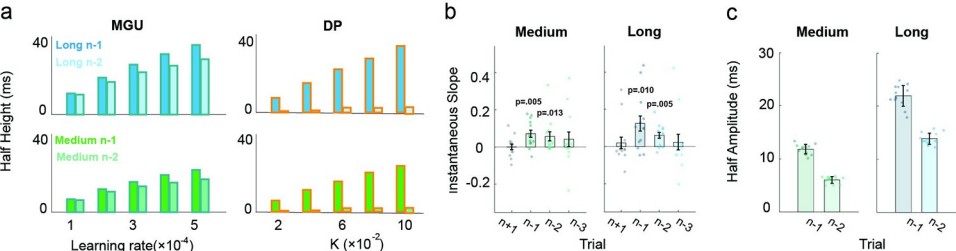

**Fig 6. The serial dependence effect becomes stronger when the range of the test distribution increases.** (a) Simulation of the half-amplitude with the MGU model (left) and the DP model(right). The upper two panels depict simulations for the long-range condition, and the lower two panels depict simulations for the medium-range condition. Simulations of the MGU model can produce robust serial dependence effects from trial n-2, whereas the DP model fails to predict this effect. (b) Instantaneous slope of the DoG-fitting curve for trials n-1, n-2, n-3, and n+1 (control condition) in the medium and long conditions. The p-values are with respect to the difference from zero. (c) Half-amplitude of the best-fitted DoG for the n-1 and n-2 trials in the medium (left) and long (right) conditions.

are consistent with the prediction of the MGU model and fail to support the DP model (Fig 6A). Moreover, the MGU is consistent with the central tendency effect in all conditions (short, medium, long), further supporting the idea that the process that produces the serial dependence effect uses local information to update the prior (S1F Fig).

We note that in the original version of the DP model, the local prior is decided by the posterior of the 1-back trial. Here we implemented a more general version of the DP model, the Dual-prior short-term memory (DPSM) model, in which the updating of the local prior after each trial incorporates information from previous trials with diminishing weight. The DPSM model generates a serial dependence from the 2-back trial (S5 Fig). However, the model predicts a near-linear serial dependence function, a prediction at odds with the data. Moreover, for the 1-back serial dependence function, the DPSM model predicts the direction of the serial dependence will flip when the current and previous stimuli are distinct from each other; this prediction is also not consistent with the data. In sum, adding a more graded working memory component to the dual prior model cannot account for the results of our experiments.

## The variance of prior increases serial dependence

The MGU and DP models also make differential predictions when the variance of the target distribution is manipulated (Fig 7A). In Experiment 4, we set the mean of the target set distribution to 900 ms (the medium condition of Experiment 3). In one condition, we created a set with low variance (720-1080ms, short-900) compared to that used in experiment 3, where the variance was larger (540-1260ms). In the second condition, we used a bimodal distribution to increase variance. The GMU predicts that the serial dependence effect will be enhanced as the variance of the target distribution is increased (Fig 7B). The logic here is similar to that described above in terms of the range of the distribution: Because a low variance test set, by definition, is more concentrated, the effect of trial-by-trial updating of the prior will be smaller relative to when the variance is high. In contrast, the DP predicts that the variance of the test set will have little effect on serial dependence (Fig 7C).

Consistent with the prediction of the MGU, the serial dependence effect was modulated by the variance of the test set. For the trial n-1 data, the best-fitted DoG in the bimodal condition ($\Delta AIC_n$ = -78.6 ± 5.6; $\Delta AIC_l$ = -18.3 ± 3.6) yielded the highest and broadest serial dependence function of the three conditions (Fig 7E); the short-900 condition (short-900: $\Delta AIC_n$ = -48.3 ± 7.8) showed the lowest and narrowest function (Fig 7D). Statistically, the half-amplitudes of the functions increased as the variance of the prior increased (short-900: 8.7 ± 0.6 ms;

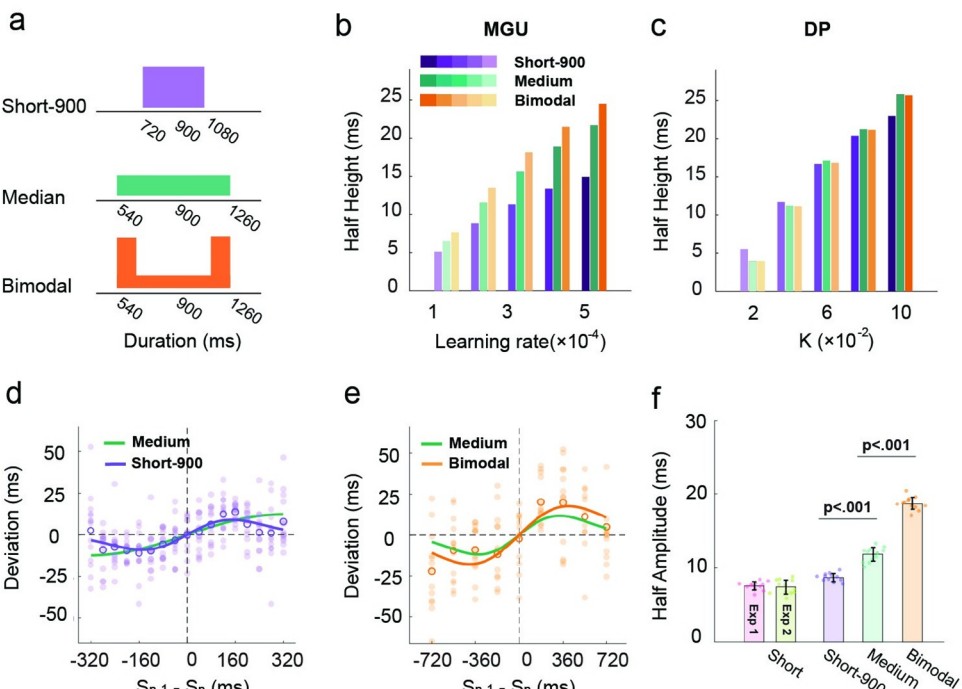

**Fig 7. Serial dependence effect becomes stronger when the variance of the test distribution increases.** (a) Illustration of the distributions of target durations in experiment 4. The short-900 condition (purple) has the same mean as the medium condition (turquoise). The bimodal (orange) has the same range as the medium condition but has larger variability. (b-c) Predicted half-amplitudes of the DoGs for the different conditions by the MGU and the DP models. (d-e) Serial dependence effect for trial n-1 of the short-900 condition (d) and bimodal condition (e). Note the scale for the x-axis is different in d and e (and thus, the function for the medium condition looks different). Each filled dot represents a participant. The open circles indicate the average across participants. The purple, orange, and turquoise lines represent the best-fitted DoG for the short-900, the bimodal-medium, and the medium conditions, respectively. Each filled dot represents one participant. Error bars represent standard error. (f) Half-amplitude of the best-fitted DoG for the trial n-1 data in Experiments 1, 2, and 4. Each filled dot represents an estimate from jackknife resampling. Error bars represent standard error.

medium: 11.8 ± 0.9 ms; bimodal: 18.7 ± 0.8ms, Zs>4.2, ps<0.001, Fig 7F). In the analysis of the individual functions, the serial dependence effect was found in both trial n-1 and trial n-2 in the bimodal condition, similar to what was found in the medium condition. In the short-900 condition, serial dependence effect was only significant from trial n-1 (S6 Fig). In sum, the results provide additional evidence that serial dependence is enhanced in higher variance test sets.

Taken together, the results from Experiments 3 and 4 are consistent with the predictions of the MGU model, and at odds with the predictions of the DP model. Manipulations of the distribution properties of the test set indicate that a unitary process gives rise to both the central tendency and serial dependence effects.

## Discussion

The internal representation of duration is context-dependent. Previous work has identified two sources of contextual bias, a central tendency bias in which the temporal representation is attracted towards the mean of a global prior, and a serial dependence effect in which recently experienced durations serve as attractors on the current representation of a stimulus duration [6, 19, 24, 27, 31]. In the current study, we asked whether the central tendency and serial dependence effects arise from a common mechanism or separate mechanisms.

We performed a series of temporal reproduction tasks in which we manipulated the distribution of the target duration. We observed central tendency and serial dependence effects in all of the experiments. Although many studies have shown a central tendency in duration perception[6, 24], the properties of a short-term contextual effect, serial dependency, is less clear. The designs and analyses used in previous work have precluded the analysis of the whole serial dependence function[11, 27]. In the current study, we consistently observed a non-linear serial dependence effect in temporal reproduction. By using a Go/No-Go task[32], we found that the effect was dependent on movement reproduction: There was an attraction effect from the previous target duration after a Go trial, but no observable effect after a No-Go trial.

To determine if long- and short-term contextual effects arise from a single or distinct Bayesian processes, we compared five computational models. We first examined two unitary models, one with a static global prior established during the initial phase of the experiment[6], and one with a local prior dictated by recent experience. Each unitary model could not account for both the observed central tendency and serial dependence effects. To generate both effects, we considered three more complex models. In one of these, we assumed that there were two separate mechanisms, one based on a global prior and one based on a local prior. For the other two models, a unitary mechanism included a process by which the global prior was continuously updated by recent experience. We rejected the unitary model in which the prior is updated in a Bayesian optimal manner since it generated a near-linear serial dependence function. In contrast, simulations of the MGU unitary and DP models yield central tendency and non-linear serial dependence effects.

To further explore the viability of the MGU and DP models, we considered predictions of the two models when we manipulated either the range or variance of the test stimuli. While both models predict that the amplitude of the serial dependence effect from the preceding trial will increase with the range of the test stimuli, only the MGU model predicts that this manipulation will also enhance the serial dependence effect from earlier trials (e.g., n-2). Furthermore, the MGU model predicts that the serial dependence effect will be enhanced when the variance of the test set is increased, even when the mean duration is fixed, whereas the DP model predicts that the variance of the prior should have negligible influence on serial dependence. The results showed that, when extending the range of the test set, the serial dependence effect was enhanced and evident in terms of the context established from the n-2 as well as the n-1 stimulus. Moreover, even we included a short-term memory component to the DP model which allows the local prior to be decided by multiple recent trials, it still fail to generate a DoG-shaped serial dependence function for the 2-back trial as what has been observed in the data. Similarly, the serial dependence effect was enhanced when increasing the variance of the test set. Together, these results provide strong support for the MGU model, indicating that the global bias and serial dependence effects arise from a unitary mechanism in which a single global prior is continuously updated after each trial by the reproduced stimulus duration.

We have presented evidence that a single prior which is dynamically updated across trials is sufficient to explain both the serial dependence and central tendency effects, as well as how they are jointly modulated by the distribution of the test set. Simple dual-prior models with a fixed global prior accompanied by a Gaussian-shaped or single-value prior determined by the most recent experience (1-back trial or short-term memory) cannot account for the behavioral results. Nonetheless, we cannot rule out that the pattern of results reported across the experiments can be captured by alternative multiple process models. One such possibility is a model that incorporates multiple processes, each adhering to the MGU model, but exhibiting distinct learning rates. However, on the grounds of parsimony, the single process MGU model provides a comprehensive account of the data, setting a point of comparison for future research.

The nonlinearity of the serial dependence function sheds light on how the prior is updated based on recent experience. Updating could follow a simple Bayesian optimal integration rule with the prior represented as a Gaussian. While this model generates a serial dependence effect, the shape of the function is approximately linear (BU, see Fig 4B). To capture a non-linear serial dependence function, we assumed that the prior is updated by increasing the weight of the Gaussian centered at the recently reproduced duration. One benefit of representing the prior with a Gaussian mixture model is that it provides a way to represent the effects of any test set, regardless of its distributional shape. Indeed, previous studies have also shown that a mixture of Gaussians provides a good fit of the internal prior in duration perception experiments [30, 36]. Here we show that an updating rule based on this assumption can account for the non-linear serial dependence function as well as how the function will shift when the variance of the test set is manipulated.

As noted above, the serial dependence effect was contingent on the participant having produced a response; it was markedly attenuated after No-Go trials. In theory, it is possible that this local effect is reflective of some sort of motor memory, where the current motor reproduction is directly biased by the previous motor reproduction, with the perceived duration unbiased. However, we think it is more likely that the serial dependence effect is mediated by perception: A prior of temporal expectation is constructed from the reproduced duration, and this prior biases the perception of subsequent target stimuli, which in turn, will be reflected in the next reproduced duration. In support of this idea, several studies have shown that making a movement of a variable duration influences performance on a subsequent duration comparison task in which no motor reproduction was made [12, 37, 38].

We note that serial dependence effects have been observed in previous studies that did not involve motor responses, including judgments of duration [13–16] or visual orientation or position [32, 33, 39].Thus, we do not claim that the absence of a serial dependence effect in the No-Go condition of Exp 2 should be taken to mean that movement is a necessary prerequisite. As with all null results, caution is warranted. It is possible that the design used in Exp 2 may have been insensitive to capture a perceptual contribution to the serial dependence effect. Alternatively, the absence of a perceptual may be specific to the duration reproduction task used in our experiments. Indeed, there is evidence from duration comparison tasks without reproduction of an attractive sequential effect [11, 40]. Follow-up experiments are needed to understand the factors that determine the weight given to perception and production in forming the prior.

We recognize that, in evaluating the five models, we assumed that a global prior was established during the initial phase of the experiment. Obviously, establishing a global prior requires some integration of the local context as the participant becomes familiar with the stimulus set. We set our initial "training" phase based on previous studies, which have shown that participants are able to generate a relatively accurate global prior of the temporal context after about 100 trials [21]. One open question is how the prior updating rate changes as a function of training. In our dynamic models, we assumed that the rate remained constant over the course of the experiment. However, it is possible that the rate of updating weakens as the prior becomes more established. Correspondingly, the serial dependence effect might become weaker over time. However, this temporal expectation system might be a rigid system that recalibrates the environment with an invariant learning rate. This characterization conforms with a conceptualization of the operation of the cerebellum in sensorimotor adaptation, and the same principles might apply to duration representation, another function associated with the cerebellum [41, 42]. The data set in the current experiments are insufficient to examine the dynamics of updating, and we see this as an important issue to be addressed in future studies.

In summary, the current study provides new insights into how context influences our sense of duration. The perception of the duration of a stimulus is sensitive to both the global distribution of the stimulus set as well as recent experience. Importantly, by examining the central tendency and serial dependence effects in a joint manner, we observed that these two forms of bias are best explained by a unitary mechanism in which a global prior is updated in an iterative manner based on each observation. Given that central tendency and serial dependence effects are ubiquitous in perception tasks [6, 18, 19, 31, 39, 43], the common Bayesian framework developed here may provide a general account of how perceptual systems adapt to environmental statistics.

## Methods

### Ethics statement

All experimental protocols were approved by the institutional review board of the School of Psychological and Cognitive Sciences, Peking University, and carried out according to the approved guidelines. Written informed consent was obtained from all participants.

### Participants

A total of sixty-four students at Peking University were recruited for the four experiments. All participants were right-handed with normal or corrected-to-normal vision. In Experiment 1, thirteen participants (8 females, mean age = 21.1, SD ± 1.0) were recruited for the two 1-hour sessions. One participant did not return for the second session. In Experiment 2, twelve participants (8 females, mean age = 24.1, SD ± 4.2) completed two 1.5-hour sessions. In Experiments 3 and 4, 52 participants (15 females, mean age = 21.1, SD ± 2.1, 13 for each of the four conditions) completed a 1-hour experiment. Participants received $10/h as compensation.

Testing was conducted in a dark room, and the stimuli were presented on a 27-inch LCD monitor (resolution of 1,024 × 768), viewed from a distance of 65 cm. The computer used a Windows 8 operating system with a refresh rate of 100 Hz. The experiment was written in MATLAB (Mathworks, Natick, MA; Psychophysics Toolbox Brainard, 1997;[44]).

### Procedure and design

**Experiment 1.** We used a "ready-set-go" time-reproduction task[6] (Fig 1A) to measure global (central tendency effect) and local (serial dependence effect) biases. Each trial started with the presentation of a gray fixation point (0.5 degrees diameter) at the center of the screen. After a random interval ranging from 0.7–1.2 s (drawn from an exponential distribution), two 100-ms stimuli flashed in sequence, with the first serving as the "Ready" signal and the second serving as the "Set" signal. The visual stimulus was either a grey ripple-shaped arc (Exp 1) or a circle (Exps 2–4) (see Fig 1) with a radius of around 12 degrees.

The interval between the onset times of the "Ready" and "Set" signals defined the target interval. Participants were instructed to press the space bar ("Go") to reproduce the target interval, with the onset of the reproduction interval defined by the "Set" stimulus. After the keypress, performance feedback was provided for 50 ms via a change in the color of the fixation point: Green indicated that the reproduced duration was within an acceptable window, and red indicated that the reproduced duration fell outside this window. The criterion window was continuously adapted based on the participant's performance such that green and red appeared with roughly equal probability. The feedback was provided to encourage the participant to pay attention to the task. It was relatively uninformative (e.g., did not provide signed

information) because we did not want participants to correct their timing based on the feedback.

Each participant completed two sessions, with each session composed of 10 blocks of 200 trials (2000 trials total). A one-minute break was provided every two blocks. The stimulus set consisted of ten target durations, ranging from 520 to 880 ms (step size of 40 ms). Target duration was randomized within a block of 100 trials with the constraint that each duration was presented 10 times.

**Experiment 2.** The goal of Experiment 2 was to evaluate whether the source of the serial dependence effect is perceptual, motoric, or a combination of both. To test this, we included Go and No-Go trials in a duration reproduction task.

After a random interval ranging from 0.7–1.2s (following an exponential distribution), a ripple-shaped stimulus was presented for the target duration. The spatial distribution of brightness was constant, but the actual shape varied across trials to avoid repetition suppression effects [45–48]. Crucially, 300 ms after the offset of the target stimulus, the fixation point changed to either a "+" or "x", with these symbols indicating that the current trial was a Go or No-Go trial, respectively. In Go trials, participants were instructed to depress the space bar for a duration that matched the target duration. The fixation point changed back to a grey circle right after the release of the keypress. On No-Go trials, participants were asked to fixate without movement until the "x" disappeared. On these trials, the grey circle reappeared after 700ms, indicating the start of a new trial. Note that no time constraints were imposed on Go trials; thus, we anticipated there would be few errors of omission (Go trials) or commission (No-Go trials). However, we assumed that the target duration would be similarly encoded on all trials since the "+" or "x" did not appear until after the target stimulus.

There were five target durations (540, 620, 700, 780, and 860 ms), with each target duration repeated on 360 trials. For each target duration, 60% were Go trials and 40% were No-Go trials. Target duration and response requirements were randomized within blocks of 180 trials. Each session consisted of five blocks (1800 total trials across the two sessions), with a one-minute break between every two blocks.

**Experiment 3.** The goal of Experiment 3 was to assess how the serial dependence effect is impacted by the range of the stimulus set. To test this, we employed two new stimulus sets: A medium condition (540, 720, 900, 1080, 1260ms) and a long condition (560, 880, 1200, 1520, 1840ms). We compared performance with these sets to the data from Experiments 1–2 (where the range was shorter, 540-860ms). The procedure in Experiment 3 was identical to that of Experiment 2, except that only Go trials were included. There were five blocks of 150/120 trials (medium/long condition), resulting in a total of 750 trials for the medium condition and 600 trials for the long condition. The difference was imposed to keep all sessions within one hour. The target duration was selected at random on each trial with the constraint that each condition occurred an equal number of times within each block.

**Experiment 4.** The goal of Experiment 4 was to examine how the serial dependence effect is impacted by the variability of the stimulus set. To test this, we employed two new stimulus sets. For the short-900 condition, the test set ranged from 720–1080 ms (steps of 40 ms, mean = 900 ms). This group has the same mean as the medium condition in Experiment 3 but with a shorter range (equal to that used in Exps 1 and 2). In this way, the variance of the test set is smaller than that of the medium condition. For the bimodal-medium condition, the test values were the same as that used in the medium condition of Experiment 3, but the extreme values (540 ms and 1160 ms) were presented three times as often as the other three test durations (670, 900, 1030ms). The short-900 condition included 10 blocks of 200 trials, and the bimodal condition included 10 blocks of 135 trials.

## Data analysis

The logic of these experiments is predicated on the assumption that participants are familiar with the temporal context. Given this, the first block of each experiment (approx. 7–10 minutes of data collection) was treated as the familiarization phase and not included in the analysis. In addition, reproduced durations shorter than 0.3 s or longer than 1.5s (2.5s for Experiments 3–4) were considered outliers and excluded from the analyses (less than 0.1% of trials).

The central tendency bias or regression to the mean was quantified as the regression coefficient between the reproduced durations and the target durations. To analyze the serial dependence effect, we used a "deviation" index. For each individual, the average reproduced duration was calculated for each target duration. The deviation was defined as the reproduced duration for a given trial minus the mean reproduced duration of all trials with that target duration [32, 33]. Positive values indicate that the reproduced duration for the present trial was longer than the average reproduction for that target and negative values indicate that the reproduced duration was less than the average reproduction.

To quantify the magnitude of the serial dependence effect, a simplified DoG curve was fit to describe the deviation index as a function of the difference between the current target duration and reference duration, where the reference could be the target duration of the previous trial (n-1), two trials back (n-2), etc., as well as following trial (n+1, serving as a baseline):

$$y = abcxe^{-(bx)^2},$$

where y is the deviation, x is the relative target duration of the previous trial, $a$ is half the peak-to-trough amplitude of the derivative-of-Gaussian, $b$ scales the width of the Gaussian derivative, and $c$ is a constant, $\sqrt{2}/e^{-0.5}$, which scales the curve to make the $a$ parameter equal to the peak amplitude. As a measure of serial dependence, we report half the peak-to-trough amplitude (half-amplitude) and half the width of the best-fitted derivative of a Gaussian. A positive value for the $a$ parameter indicates a perceptual bias toward the target durations of the previous trials. A negative value for the $a$ parameter indicates a perceptual bias away from the target durations of the previous trials.

We fit the Gaussian derivative at the group and individual level using constrained nonlinear minimization of the residual sum of squares. Jackknife resampling was applied to estimate the variation of the parameters for the group-level fit, where each participant was systematically left out from the pooled sample. The standard deviation of those estimates represented the standard error of the parameter at the group level. The half-amplitude and half-width of the best-fitted DoG were compared between groups with a t-test, where the t-value was computed with the mean and the variance of the parameters estimated from the jackknife resampling procedure. Bonferroni correction was applied for multiple comparisons. To test the goodness-of-fit of our model, we computed the ΔAIC for the DoG model compared with either a non-model (y = 0, $\Delta AIC_n$) or a linear model (y = $kx$, $\Delta AIC_l$). A negative ΔAIC indicates DoG performed better than the alternative models.

To determine the extent of serial dependency, we fitted individual serial dependence functions in which the reference could be the target duration of the previous trial (n-1), two trials back (n-2), etc. We also tested the serial dependence function for the n+1 trial. Given that this reference stimulus has not been experienced, there should be no serial dependence effect here, providing a test of whether the deviation measure is a valid index to analyze serial dependence. Since individual serial dependence functions do not always show a strong nonlinearity, $a$ parameter (half-amplitude) can become unreasonably large. Thus, we opted to use the instantaneous slope of the DoG at the inflection point (*abc*) when estimating the presence of a serial

dependence effect at the individual level. Note that this index measures the sign of the serial dependence independent of whether or not the function is linear.

In Experiment 2, the Go trials were sorted into two groups based on whether the reproduced interval on that trial was preceded by a Go trial or a No-Go trial. We calculated the average reproduced durations and the deviation using the same protocol as Experiment 1 and then fit DoG functions separately for the Go/Go trial sequence and the No-Go/Go trial sequence to measure the serial dependence effect from trial n-1. For each function, we calculated the instantaneous slope of the DoG at the inflection point. Note that if the serial dependence effect is dependent on a motor response, there should be no serial dependence effect when the preceding trial was a No-Go trial. For Go trials preceded by a No-Go trial, we also analyzed the serial dependence effect from n-2 Go trials. Group-level DoG fitting was performed on n-1 and n-2 Go trials separately to quantify the amplitude of serial dependence.

One-sample t-tests and paired t-tests were applied at the group level for comparisons. Normality and equal variance assumptions were assessed prior to the t-tests. The Wilcoxon Sign-rank test was applied when the normality assumption was violated. Two-tailed P values are reported for all statistic tests, and the significance level was set as $p < 0.05$. All analyses were performed with MATLAB 2018b (The MathWorks, Natick, MA).

## Models

To account for central tendency and serial dependence effects, we implemented five Bayesian models: Two single process models (Global-only, Local-Only), a model with both a static global prior and a local prior (Dual-prior), and two unitary models that capture how a global prior is dynamically updated by recent experience (Bayes-Updating & Mix-Gaussian-Updating).

## Global-Only model

This model is based on a Bayesian observer model that uses a Bayes-Least-square as the mapping rule [6]. We assume that the observer builds up an internal prior, $\pi(t_s)$ based on the target durations ($t_s$) observed during an initial exposure phase, and subsequent judgments are made by reference to this static prior. The likelihood function describes the probability of the perceived duration ($t_p$) given $t_s$.

$$p(t_p|t_s) = N(t_p|t_s, v_p) \tag{1}$$

where $N(x|m, s)$ represents a normal distribution with mean m and standard deviation s, and $v_p$ scales the sensory noise. The posterior, $\pi(t_s|t_p)$, is the product of the prior multiplied by the likelihood function and appropriately normalized.

$$\pi\left(t_s|t_p\right) = \frac{p(t_p|t_s)\,\pi(t_s)}{\int p(t_p|t_s)\,\pi(t_s)dt_s} \tag{2}$$

The loss function, $l(t_e, t_s)$, was used to convert the posterior into a single estimate, $t_e$, the mean of the posterior in the present situation.

$$l(t_e, t_s) = (t_e - t_s)^2 \tag{3}$$

$$t_e = \underset{t_e}{\operatorname{argmin}}\left[\int l(t_e, t_s) * \pi(t_s t_p)dt_s\right] \tag{4}$$

The Bayesian observer makes a response based on $t_e$:

$$p(t_r|t_e) = N(t_r|t_e, v_m) \tag{5}$$

where $t_r$ is the reproduced duration, and $v_m$ scales the motor noise. Previous studies [6, 9] have shown that scalar forms of perceptual and motor noise provide a better fit of the behavior compared to when $v_p$ and $v_m$ are treated as constants. Thus, we set $v_p$ as $n_p{}^*t_s$, and $v_m = n_m{}^*t_e$, where $n_p$ and $n_m$ are constants.

For this baseline, Global-only model, we simulated the results under the assumption that the prior was established based on a data set that would be experienced during the first 7–10 minutes of the experiments, and then remained fixed for the duration of the experiment. We assumed that participants learn the true distribution of target durations as the prior:

$$\pi(t_s) = U(t_s, [500\ ms, 900\ ms]) \tag{6}$$

where $U(x, [y, z])$ represents a uniform distribution ranging from $y$ to $z$.

## Local-Only model

To capture the serial dependence effect, we adapted a Bayesian integration model from a previous study that examined serial dependence in magnitude estimation [26]. In the current context, this model assumes that the observer integrates the previous response ($t_r$) with the current stimulus ($t_s$) when estimating the duration of the current stimulus:

$$t_{e,i} = (1 - W_{i-1})t_{s,i} + W_{i-1}t_{r,i-1} \tag{7}$$

where $t_{e,i}$ and $t_{s,i}$ indicate the $t_e$ and $t_s$ of trial i, and $t_{r,i-1}$ is the reproduced duration of trial i-1. $W_{i-1}$ is the weight the observer assigns to the trial n-1 response when estimating the current duration. Following Bayes rule to integrate the two samples dictating the current percept, the observer specifies $W_{i-1}$ as

$$W_{i-1} = \frac{1/\sigma_{i-1}^2}{1/\sigma_i^2 + 1/\sigma_{i-1}^2} = \frac{\sigma_i^2}{\sigma_i^2 + \sigma_{i-1}^2} \tag{8}$$

where $\sigma$ is the variance of the estimate. The weight is influenced by the distance between the two stimuli,

$$W_{i-1} = \frac{\sigma_i^2}{\sigma_i^2 + \sigma_{i-1}^2 + \left(t_{s,i} - t_{s,i-1}\right)^2} \tag{9}$$

The variance is assumed to follow a power law.

$$\sigma = Kt_s{}^\alpha \tag{10}$$

Assuming that time perception follows a scalar rule, $\alpha = 1$. $K$ is a free parameter regulating the behavior of the model.

## Dual Prior model (DP)

We combined the Local-Only and Global-Only models to create a Dual Prior model. It contained two Bayesian processes. For the first integration, the observer integrates the current stimulus ($t_s$) with the prior $\pi(t_s)$ to get an estimation $t_e'$, following Global-Only model (Eqs [4]–[7]). The second integration estimates $t_{e,i}$ based on $t_{r,i-1}$, $t_{e,i}'$ and $t_{e,i-1}'$ following the Local-

Only model. The Formulas [7] and [9] are rewritten as

$$t_{e,i} = (1 - W_{i-1})t_{e,i}' + W_{i-1}t_{r,i-1} \tag{11}$$

$$W_{i-1} = \frac{\sigma_i^2}{\sigma_i^2 + \sigma_{i-1}^2 + (t_{e,i}' - t_{e,i-1}')^2} \tag{12}$$

## Duel Prior Short-term Memory model (DPSM)

In the DP model, we assume the local prior is determined by the 1-back reproduction. Here, we introduce a more general version of this model that assumes the local prior is decided by short-term memory. Specifically, the local prior ($t_{prior}$) is updated after each reproduction following a Kalman-filter:

$$t_{prior,i+1} = lt_{r,i} + (1 - l)t_{prior,i} \tag{13}$$

where $l$ is the learning rate that decides how fast the prior is updated. Eq [7] can be rewrite as

$$t_{e,i} = (1 - W_{i-1})t_{s,i} + W_{i-1}t_{prior,i} \tag{14}$$

## Bayes Updating model (BU)

We also considered two unitary process models in which a global prior is updated in a dynamic manner. For the Bayes Updating model, the prior is updated following Bayes rule after each observation. Since the prior and likelihood are Gaussian, this model can be expressed as a Kalman filter. On each trial, the estimated duration ($t_e$) is a weighted sum of $t_o$ and the duration of the stimulus ($t_s$).

$$t_e = (1 - w)*t_o + w*t_s \tag{15}$$

where the weight $w$ is determined by the sensory noise and variance of the prior. As such, as $w$ increases, the weight given to the prior will increase (i.e., attraction to central tendency). Participants make a motor response based on $t_e$ with Gaussian motor noise (see Eq [5]). Given that Experiment 2 showed that serial dependence is primarily induced by the reproduction component rather than the perceptual component, we assumed the prior is updated according to $t_r$. After the motor response, $t_0$ is updated based on $t_r$ following another Kalman filter:

$$t_o' = (1 - k)*t_o + k*t_r \tag{16}$$

where $t_o'$ is the new reference point for the next trial, and $k$ represents the learning rate. To address the scalar property of timing noise (Weber law), $t_r, t_s, t_e,$ and $t_o$ are taken to be the log value of the respective durations. Note, that an alternative version of this model could have the observer directly use the posterior as a new prior (what is known as a fully iterative model). However, as with the Local-Only model, a fully iterative model will largely overestimate serial dependence given a reasonable central tendency effect.

## Mixed-Gaussian Updating model (MGU)

In a second unitary model, the prior is represented as a Gaussian mixture model. This model is identical to the Global-Only model (Eqs [1–5]) except that the prior is updated after each trial to generate a better estimate of the temporal context. We applied a simple updating rule here, in which the observer adds a normal distribution centered at $t_r$ with a standard deviation

of $v_p$ to the old test set. The posterior is then appropriately normalized:

$$\pi'(t_s) = \frac{\pi(t_s) + r*N(t_s|t_r, v_p)}{\int [\pi(t_s) + r*N(t_s|t_r, v_p)]dt_s} \tag{17}$$

where $\pi'(t_s)$ is the new test set, and $r$ is the learning rate.

## Simulation procedure

Simulations of each model were conducted to evaluate the results of Experiment 1. The data from 100 pseudo-participants were generated for each simulation. To determine the values of the free parameters for the simulations, we referred to a previous study that used a similar design to that employed here and evaluated the results with the BLS model, or what we refer to as the Global-Only model [6]. Based on their results, we set $n_p = 0.10$ and $n_m = 0.06$ for the Global-Only, MGU, and DP models. Other parameters were determined to make the amplitude of the serial dependence function roughly similar to the behavioral results. Specifically, we set $w = 0.7$ and $k = 0.3$ for the BU model, the learning rate $r = 0.3$ for the MGU model, and $K = 0.06$ for the Local-Only and DP models. We designed Experiments 3 and 4 to focus on the MGU and DP models, the two models that produce both central tendency and non-linear serial dependence effects. For simulations of the short conditions, the step between every two adjacent stimuli was the same as what was used in the other experiments (40 ms). For simulations of the medium and the long conditions, the step size was set to a smaller value (70 ms) than used in the experiments to improve resolution. The parameters $n_p$ and $n_m$ were fixed at the values used in the previous simulations, while a series of $r$ and $K$ values were tested.

## Supporting information

**S1 Fig. Central tendency of the reproduced duration as predicted by the MGU model in all experiments.** (a-e) Reproduced duration is plotted as a function of target duration for Experiments 1, 3, & 4. The shaded area indicates S.E. The median slope ± S.E. is reported on each figure. (f) The predicated slopes for the central tendency of the MGU model provide a good fit to the data. The dots and error bars indicate median slope ± S.E.
(TIF)

**S2 Fig. Histogram of the difference between the previous target duration ($S_{n-1}$) and the current target duration ($S_n$).** For individual participants, there are few trials with a large difference between stimuli, especially for experiment 1 and the short-900 condition. As such, the DoG may not provide a good fit when used to estimate serial dependence curves at the individual level.
(TIF)

**S3 Fig. Deviation index functions for Experiment 2.** (a) A prominent DoG curve can be seen for the n-1 data when trial n-1 was a Go trial. (b) This curve is markedly attenuated when trial n-1 was a no-Go trial. (c) A small DoG curve is evident for the n-2 trial when n-1 was a No-Go trail and n-2 is a Go trial. The thick dashed line is the best-fitted DoG curve. Shaded areas indicate standard error.
(TIF)

**S4 Fig. A local-only model cannot explain the results in Exp 1 across a range of parameter values.** (a) Predicted relationship between central tendency and serial dependence for the local-only models. The depicted function shows the prediction of the model as the value of K is manipulated. The gray bar indicates the data from Exp 1, with the width of the bars

indicating SE. (Note that the SE of half amplitude is very small). (b-c) Prediction of the central tendency and serial dependence effects for models with a K value of either 0.31 or 0.06. The model cannot capture both effects simultaneously.
(TIF)

**S5 Fig. A dual-prior model with short-term memory (DPSM) fails to predict the non-linear serial dependence.** (a) Serial dependence function of the 1-back trial in the medium (top row) and long (bottom row) conditions of Exp 3. Thick line indicates the best-fitted DoG function. (b-c) 1-back serial dependence function predicted by the DPSM model with different values of the L (b) and k (c) parameters. In all simulations, the DPSM fails to generate a DoG-shaped serial dependence function for the 1-back trial. (d) Serial dependence function of the 2-back trial. Thick line indicates the best-fitted DoG function. (e-f) 2-back serial dependence function predicted by the MGU and DPSM models. The DPSM model predicts a monotonic, near-linear serial dependence function for the 2-back trial. Error bars indicate standard error.
(TIF)

**S6 Fig.** Instantaneous slope of the fitted DoG functions for trials n-1, n-2, and n-3, and future (n+1) trials in the short-900 (a) and bimodal conditions (b). A significant serial dependence effect can be observed from the n-2 trial in the bimodal condition rather than the short-900 condition in experiment 4. Dots indicate individual data points and error bars represent standard error. The p-values are based on a test of whether the observed values differ from zero.
(TIF)

## Acknowledgments

We thank Linfeng Han, Jiashu Wang, and Huihui Zhang for valuable discussions.

## Author Contributions

**Conceptualization:** Tianhe Wang.

**Data curation:** Tianhe Wang, Yingrui Luo.

**Formal analysis:** Tianhe Wang.

**Funding acquisition:** Tianhe Wang, Yan Bao.

**Visualization:** Tianhe Wang.

**Writing – original draft:** Tianhe Wang.

**Writing – review & editing:** Tianhe Wang, Yingrui Luo, Richard B. Ivry, Jonathan S. Tsay, Ernst Pöppel, Yan Bao.

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
