## [Decision Letter · Decision Letter 0]

22 Feb 2023

Dear Mr Wang,

Thank you very much for submitting your manuscript "A Unitary Mechanism Underlies Adaptation to Both Local and Global Environmental Statistics in Time Perception" for consideration at PLOS Computational Biology.

As with all papers reviewed by the journal, your manuscript was reviewed by members of the editorial board and by several independent reviewers. In light of the reviews (below this email), we would like to invite the resubmission of a significantly-revised version that takes into account the reviewers' comments.

We cannot make any decision about publication until we have seen the revised manuscript and your response to the reviewers' comments. Your revised manuscript is also likely to be sent to reviewers for further evaluation.

Sincerely,

Roland W. Fleming, PhD

Academic Editor

PLOS Computational Biology

Thomas Serre

Section Editor

PLOS Computational Biology

Reviewer's Responses to Questions

**Comments to the Authors:**

Reviewer #1: The present manuscript by Wang et al. investigates the computational formulation of two hysteretic phenomena in time perception, central tendency and serial dependence. By comparing the predictions from various Bayesian models with the behavioral data from time reproduction experiments, the authors claim that the two phenomena stem from a unitary mechanism that updates a global prior locally after each trial.

One central focus of the study is how to computationally implement the feature tuning (termed "non-linearity" in the manuscript) of serial dependence, providing a clear distinction from existing models. This is clever, and surely a valuable addition to the field. Another key aspect addressed is the question of whether global and local priors are regulated by the same or distinct mechanisms. The authors evaluate two models (Mixed Gaussian Updating model and Dual Priors model) that can generate both central tendency and serial dependence, based on the temporal decay of serial dependence. While they have made efforts to keep these models simple and straightforward, as a result, some assumptions are arbitrary and potentially inconsistent with previous studies.

Although I am inclined towards the MGU model, the DP model requires significant modifications to ensure a fair comparison. The discussion and conclusion should be reevaluated after the revisions suggested below. Regardless, the present study constitutes a valuable contribution to the field and is likely to be of broad interest to readers of PLOS Computational Biology.

Major comments:

1. First and foremost, the serial dependence in duration perception has been reported (e.g., Togoli et al., 2021; Wiener et al., 2014; Wiener & Thompson, 2015), which should be acknowledged accordingly. Moreover, Zimmermann & Cicchini (2020) explicitly claims that such a contextual effect in time perception occurs at the perceptual level. The authors already discuss this perception vs. movement/response topic in a very balanced manner (ll.443-450), which is great, but mostly based on serial dependencies in other domains. I would suggest citing the papers above in the same paragraph.

As suggested by Kiyonaga et al. (2017) and further supported by many subsequent studies, there are many forms of serial dependence. Serial dependencies are stimulus- and/or task- and/or modality-specific. Thus, as one possibility, the serial dependence induced by time reproduction may be specific to motor timing.

2. The models and figures require substantial modifications. Please see below for details.

2-1. Figure 3

The flow chart and color codes are somewhat complicated. Also it appears that the DP and Local-Only models yield the same prediction, which is actually not the case.

(Panel a) The "perception noise" should be rephrased into "sensory noise" or something similar, as the term "perception noise" has an assumption that sensory likelihood IS perception. That is, it assumes that such a Bayesian integration happens after perceptual processes, contradicting recent studies (e.g., Cicchini, Benedetto, Burr, 2021; Murai & Whitney, 2021; etc.).

(Panel b) The term "motor noise" was confusing until reading the Models section. It first appeared to me as if the response was directly drawn from the posterior and the posterior width is called motor noise. Could the authors somehow modify the figure or elaborate in the legend so it reflects what they actually did (convert the posterior to a single value [eq.3-4] and then introduce a scalar motor noise component [eq.5])?

2-2. Figure 4

(Panel a) The Global-Only model predicts only a tiny central tendency compared to the observed effect size. However, many studies have successfully predicted the central tendency with this model, including Jazayeri & Shadlen's seminal work and many subsequent papers (e.g., Cicchini et al., 2014 JNS; Karaminis et al., 2016; Martin et al., 2017; Murai & Yotsumoto, 2018; etc.). This is presumably because np and nm values selected in the simulation (l.732) were not suitable for the present dataset.

(Panel b) The Local-Only model and the DP model show quite a narrow feature tuning, indicating a similar problem in parameter selection.

(Panel d) The prior(n-1) suddenly appears without any explanations as a uniform distribution spanning over 500-1900 ms, which is totally different from stimulus distribution. I guess this is partly because otherwise the posterior would be completely flat in the right panel, but this visualization is very confusing. To be clear, I'm NOT against the concept itself. Just the visualization.

2-3. Figure 5

This figure and underlying models should be overhauled. To be fair, I personally advocate the MGU, and agree with the conclusion itself. However, I'm not fully sure if the behavioral data in the present study is really able to distinguish the MGU and the DP.

The most serious concern is that the DP has a very strong and arbitrary assumption that the local prior depends only on reproduction(n-1). If the local prior is derived from short/working memory trace from previous trials, as suggested by recent studies, the local prior could extend in time over a couple of trials. This should yield similar predictions about SD(n-2) as the MGU. Would it be possible to implement such a temporal component to the DP model? It might be better to drop the n-2 arguments and Figure 6 unless the authors present another point that MGU and DP would yield distinct predictions.

Furthermore, could the authors elaborate why the global prior is first integrated with the likelihood and then the local prior in the DP model? My hunch is the opposite. Would the prediction change if the order is flipped?

(Panel d) I understand this as a computational concept, but obviously human observers cannot retain such a complex prior like the prior(n) in the short condition (see Acerbi et al., 2014). I suppose this will hold even with simpler prior shapes, though. The same logic is used for the variance argument (Figure 7), but it's less intuitive.

(Panel e; ll.277-279) The sensory likelihood should follow Weber's law both in MGU and DP (and any other models). Why should this predict the range-dependency of serial dependence only for DP?

In any case, the novelty and strength of this study is that no existing models (Global prior, Kalman filter, etc.) can simultaneously implement central tendency and serial dependence including its tuning properties. I'm not sure if it's a good idea to dip into the debate about single vs. multiple mechanisms (it's certainly a quite important issue though!).

Minor comments:

3. (citation on l.43) Both the central tendency and the serial dependence typically happen between discrete events, as in the present study. While Fischer paper is in line with this, Manassi paper demonstrates a contextual effect WITHIN a continuous event, which is wildly different from the focus of this study. Although Manassi paper is an excellent manifestation of a new form of serial dependence, I believe there should be more suitable papers to cite here. I would suggest the papers listed in comment #1.

4. (citation on l.49) I would suggest citing another relevant paper about Bayesian calibration of timing perception (Miyazaki et al., 2006, Nat. Neurosci.).

5. (citation for Kalman-filter model) In addition to the Internal Reference model by Dyjas and colleagues, Kalman(-like) filter has been used to model serial dependence and central tendency (e.g., Cicchini et al. 2014; Taubert et al., 2016; Petzschner & Glasauer, 2011).

5. (l.142) "the serial dependence effect is about 8%"

While a few papers have tried to quantify serial dependence by height/width, most readers would not understand what the authors mean without explanation about how and why they chose to calculate the effect size this way. From another perspective, considering that the mean stimulus duration was 700 ms, one would argue that the half-height of ~7ms is just about 1% of stimulus duration.

6. (ll.156-159) In Figure S3, n-1 no-Go trials seem to show positive serial dependence. Although only the instantaneous slope result is indicated in Fig.2c, was the half height also not significant? Also, perhaps the x-axis label should be S(n-2) - S(n) for the rightmost panel.

7. Some typos:

(l.249) "integrated with a static global prior, generating *serial dependence*" -> maybe central tendency?

(l.683) "The formulas [11] and [13]" -> [7] and [9]

(Figure 2c, tick numbers on y-axis) 1 -> 10; -1 -> -10

Reviewer #2: The authors study temporal reproduction. In this task typical a central tendency and serial dependencies of reproduced intervals are observed. The authors test 3 different models that can account for both phenomena plus 2 specific models. 3 of the models can be rejected based on the first two experiments. The remaining two models are nicely pitted against each other by 2 targeted further experiments. Based on their data the authors argue for a “unitary mechanism model”, where a global prior is updated by adding local priors.

The question is of interest, the methods appear sound, experiments, modelling and model tests are carefully done. The manuscript is not easy to read, but this is also due to the topic chosen. I have mainly one concern. In my view the authors should not talk about “unitary mechanism” vs. “dual prior” models. According to my understanding one crucial difference between the “unitary” MGU and the “dual” DP model is that the former stores local information about all previous trials, whereas the latter stores local information only about the last trial (plus global info). This point should be highlighted in the discussion, while the claim of a “unitary” vs. different mechanisms should be downplayed. Also, I am pretty sure that a “dual prior” model covering more of the immediately previous info in its local prior could fit the results as well as does the current MGU. Except for that I recommend acceptance.

“sense of passage of time”: This concept should not be combined with measuring duration reproduction, because in the timing literature different measures and results are associated with “passage of time” and “time estimation”. So, please do not use this phrase in the present context.

Fig. 2 C, There are mistakes in the y-axis labels.

Reviewer #3: In this manuscript, the authors investigated the interplay between serial dependence and central tendency effects. To this purpose, they investigated timing perception as a function of recent experience and the global distribution of the stimulus across trials. By running behavioural experiments and testing a variety of models, they proposed that a model based on an update of a global prior trial-by-trial could explain the results.

I think the manuscript is pretty well written and interesting. However, I have some major and minor concerns on the main message of the manuscript (see below).

Major comments:

I believe the authors should be careful with arguing that serial dependence and central tendency originate from the same mechanism. First, showing that a model can account for both effect does not necessarily implies a common mechanism. Second, and more importantly, serial dependence has some very peculiar characteristics (tunings) which cannot be explained by simple central tendency biases. For example, spatial tuning (ref 1-3) implies a difference in serial dependence independently on response reproduction or decision. In addition, serial dependence is gated by attention (ref 1). Can the unitary model explain these characteristics of serial dependence? The authors should test whether their model can account for the tuning properties of serial dependence.

Regarding Experiment 2, there is a conflicting literature on the role of previous decision/attention on serial dependence. Some studies have found that serial dependence does not necessarily depend on decision (ref 1,4), whereas others did (see Pascucci et al. 2019). I saw the authors acknowledged some of these findings, but it would be useful to rephrase statements like "This dissociation suggests that serial dependence in timing mainly arises from sequential effects associated with temporal reproduction". Several factors may be at play here, such as attention, decision, memory, motor response etc etc.

Minor comments:

Line 76: "the serial dependence effect has not been well established in timing." There is a large body of literature of serial dependence in time perception (see 5-10 below). The authors should rephrase.

Figure 2 (and across the manuscript). The authors refer to "height" as the half amplitude of the DoG curve. Across the literature, this measure is called "half amplitude". It would be great if the authors could rephrase this to be consistent with the literature.

References:

1 Fischer, Jason, and David Whitney. "Serial dependence in visual perception." Nature neuroscience 17.5 (2014): 738-743.

2 Manassi, Mauro, Árni Kristjánsson, and David Whitney. "Serial dependence in a simulated clinical visual search task." Scientific reports 9.1 (2019): 19937.

3 Manassi, Mauro, et al. "Serial dependence in the perceptual judgments of radiologists." Cognitive research: principles and implications 6 (2021): 1-13.

4 Murai, Yuki, and David Whitney. "Serial dependence revealed in history-dependent perceptual templates." Current Biology 31.14 (2021): 3185-3191.

5 Roseboom, Warrick. "Serial dependence in timing perception." Journal of experimental psychology: human perception and performance 45.1 (2019): 100.

6 Recio, Renan Schiavolin, et al. "Dissociating the sequential dependency of subjective temporal order from subjective simultaneity." Plos one 14.10 (2019): e0223184.

7 Bilacchi, Cristiano Moraes, et al. "Temporal dynamics of implicit memory underlying serial dependence." Memory & Cognition 50.2 (2022): 449-458.

8 de Azevedo Neto, Raymundo Machado, and Andreas Bartels. "Disrupting short-term memory maintenance in premotor cortex affects serial dependence in visuomotor integration." Journal of Neuroscience 41.45 (2021): 9392-9402.

9 Motala, Aysha, Huihui Zhang, and David Alais. "Auditory rate perception displays a positive serial dependence." i-Perception 11.6 (2020): 2041669520982311.

10 Zimmermann, Eckart, and Guido Marco Cicchini. "Temporal context affects interval timing at the perceptual level." Scientific Reports 10.1 (2020): 8767.

**Have the authors made all data and (if applicable) computational code underlying the findings in their manuscript fully available?**

Reviewer #1: Yes

Reviewer #2: None

Reviewer #3: Yes

PLOS authors have the option to publish the peer review history of their article (what does this mean?). If published, this will include your full peer review and any attached files.

Reviewer #1: No

Reviewer #2: No

Reviewer #3: No
---

## [Editor Report · Decision Letter 1]

19 Apr 2023

Dear Mr Wang,

We are pleased to inform you that your manuscript 'A Unitary Mechanism Underlies Adaptation to Both Local and Global Environmental Statistics in Time Perception' has been provisionally accepted for publication in PLOS Computational Biology.

Best regards,

Roland W. Fleming, PhD

Academic Editor

PLOS Computational Biology

Thomas Serre

Section Editor

PLOS Computational Biology

---

## [Editor Report · Acceptance letter]

2 May 2023

PCOMPBIOL-D-22-01918R1 

A Unitary Mechanism Underlies Adaptation to Both Local and Global Environmental Statistics in Time Perception

Dear Dr Wang,

I am pleased to inform you that your manuscript has been formally accepted for publication in PLOS Computational Biology. Your manuscript is now with our production department and you will be notified of the publication date in due course.

With kind regards,

Zsofia Freund
